# A subnational socioeconomic assessment of family planning levels, projections, and disparities among married women of reproductive age in Cameroon

**Raïssa Shiyghan Nsashiyi**[1]*, **Md Mizanur Rahman**[2], **Lawrence Monah Ndam**[1,3], **Masahiro Hashizume**[4]

**1** Institute for Nature, Health, and Agricultural Research (INHAR), Buea, Cameroon, **2** Research Centre for Health Policy and Economics, Hitotsubashi Institute for Advanced Study, Hitotsubashi University, Tokyo, Japan, **3** Agroecology Laboratory, Department of Agronomic and Applied Molecular Sciences, Faculty of Agriculture and Veterinary Medicine, University of Buea, Buea, Cameroon, **4** Department of Global Health Policy, Graduate School of Medicine, The University of Tokyo, Tokyo, Japan

\* raissa.nsashiyi@gmail.com

## Abstract

### Background

Local-level socioeconomic gradients significantly influence access to reproductive health services in developing countries. This study examines disparities in family planning use among married women of reproductive age across Cameroon's subnational territories, highlighting inequities often overlooked in national analyses. Furthermore, it incorporates HIV status (a key yet frequently omitted covariate) into the assessment of family planning determinants.

### Methods

A Bayesian hierarchical model incorporated with Cameroon Demographic and Health Survey cross-sectional data (between 1991 and 2018) was employed to generate estimates of family planning indicators per residence, wealth, and education categories within each region. Slope index of inequality was used to quantify disparities. The determinants analysis involved Bayesian logistic regression.

### Results

Estimates for 2023 revealed that the *Centre* region's urban and rural areas had the highest modern contraceptive prevalence rate overall, with 49.0% (24.9–73.8) and 28.2% (12.4–52.3), respectively. The rural *Far North* had the least estimate [3.9% (1.5–10.5)]. Demand satisfied with modern methods was highest among *Adamawa* region's richest quintile [82.9% (58.1 to 94.4)] and higher educated [85.9% (69.5 to 94.2)], and lowest among the *East* region's poorest [5.3% (1.5 to 16.5)] and *Far North*'s none-educated [8.6% (3.3 to 20.4)]. Unmet need for modern methods was lowest among the *West* region's richest [5.1% (1.8 to 13.5)] and highest among the *Littoral*'s poorest [23.1% (9.4 to 47.4)]. 2030

**Data availability statement:** The data used in this study is freely available upon request at the Demographic and Health Survey Program website: https://dhsprogram.com/data/available-datasets.cfm?ctryid=4

**Funding:** The author(s) received no specific funding for this work.

**Competing interests:** The authors have declared that no competing interests exist.

projections show the widest wealth- and education-based gaps for demand satisfied with modern methods in the *Adamawa* [27.0 percentage points (%p) (2.3 to 51.6) and 79.3%p (73.9 to 84.7), respectively]. Age ≥ 20 years, higher education level, practising Catholic/Christian religion, having ≥ one living child(ren), and higher household wealth quintile, were associated with increased odds of modern contraceptive use.

## Conclusion

Increased focus is essential on rural, poorer, and less educated populations, particularly in the Northern regions, to effectively address family planning inequities across Cameroon.

## Introduction

In the context of the Sustainable Development Goals (SDGs), particularly SDG 3.7.1, which emphasizes the importance of demand for family planning satisfied with modern methods [1], it is crucial that current assessments reflect progress relating to inequities that distinguished levels in the past [2]. Despite renewed efforts [3] and significant gains made in the expansion of access to basic family planning services and commodities across low- and middle-income countries (LMICs) [4,5], evidence continues to mount of substantial disparities that are defined by varied socioeconomic characteristics such as wealth, education, and geographic location [2,6]. Research indicates that disadvantaged socioeconomic groups continue to experience lower coverage of family planning services [2,6], which correlates with their higher risks of adverse health outcomes, including maternal and child mortality, that these services are designed to mitigate [7].

Evidence from LMICs shows lower levels of use of [8] and demand satisfied with modern contraceptive methods [9] among poorer, uneducated, and rural populations of women who are married or in a union. In Cameroon specifically, the gaps in demand satisfied with modern methods based on wealth, education, and area of residence have not only persisted but also widened by at least 16.3, 15.6, and 8.2%, respectively, between 1991 and 2011 [6]. While there has been an overall increase in modern contraceptive use [5,10] amidst increased investments toward expanding access to family planning services nationwide [11], these disparities underscore the necessity for further investigation to inform specific strategies that address the unique needs of different subpopulations. Previous studies conducted in LMICs have predominantly focused on national-level socioeconomic disparities in family planning use [6,12–15]. However, further disaggregated analyses by smaller areas can provide insights into localized needs and enhance the implementation of programmes per subpopulation needs. In Cameroon, conducting subnational socioeconomic assessments of family planning can inform policies and facilitate targeted interventions for disadvantaged groups, at lower levels that align with the country's administratively organized healthcare delivery system [16].

Meanwhile, a critical gap exists in the assessment of family planning determinants, particularly regarding the inclusion of HIV status as a covariate. Studies conducted across sub-Saharan African (SSA) countries often omit this variable [17–23], despite the region having the highest HIV rates globally [24], and infection status significantly influencing the use of common barrier contraceptives like male/female condoms [25,26]. Beyond their role in family planning for preventing unintended pregnancies, condoms also offer protection against the transmission of HIV and other sexually transmitted infections [25,26]. Moreover, prior research has demonstrated that HIV-positive status is a significant predictor of family planning practices overall [27].

This study, therefore, aims to explore levels of and disparities in modern contraceptive use, unmet need for, and demand satisfied with modern family planning methods based on area of residence, wealth, and education across Cameroon's 10 regions up to 2030. Additionally, it examines individual- and -community-level factors that differentiate the use and non-use of modern family planning methods on a national scale are examined.

## Methods

### Data

Data were sourced from five Demographic Health Survey (DHS) cross-sectional surveys conducted in Cameroon in 1991, 1998, 2004, 2011, and 2018. Sample sizes ranged between 3,871 (DHS-1991) and 15,426 (DHS-2011). For area of residence-, wealth-, education-based analysis of either family planning indicators, totals of 108, 273, and 213 survey observations were incorporated (Supplementary material, S1 Table).

### Outcome variables

Outcome variables were defined in line with previous studies [5,28,29]. Modern contraceptive prevalence rate equals the percentage of women who are or whose husband/partners are currently using any modern contraceptive method. Unmet need for modern methods is the percentage of women who are not currently using any method of contraception to prevent pregnancy but want to space or limit childbearing. This indicator also includes women using traditional family planning methods, as they are considered to have an unmet need for more effective modern contraceptive methods. Lastly, demand satisfied with modern methods shows the percentage of women who want to space or limit childbearing and who use a modern contraceptive method. Detailed calculations are presented as supplementary material (S2 Table). Modern contraceptive methods include sterilisations, oral contraceptive pills, intrauterine devices, injectables, implants, condoms, lactational amenorrhea method, standard days method, emergency contraception, and vaginal barrier methods [30]. Traditional methods of contraception include abstinence, withdrawal (coitus interruptus), the rhythm method (calendar method), douching, and other folk methods [5].

### Predictor variables

*Trends analysis*: Year-specific Human Development Index (HDI) [31] for each region was used. The HDI is a summary measure of average achievement in key dimensions of human development, including life expectancy, education, and per capita income [32]. HDIs were projected up to 2030 via linear extrapolation.

*Determinants analysis*: Following preceding literature [17–21,33] individual-level variables, i.e., age, level of education, religion, number of living children, media exposure, and wealth quintile, as well as community-level variables, i.e., area of residence and region were incorporated. Additionally, HIV status was included as an individual-level predictor [27]. Both the disparities and determinants assessments represent estimates for married women of reproductive age, i.e., women aged 15 to 49 years who are married or in a union.

### Statistical analysis

Each outcome variable was logit transformed to ensure that the modelled estimates remained within the range of 0 to 100%. Three separate Bayesian models [34] were fitted such that estimates of each family planning indicator would be dependent on the corresponding HDIs. These were incorporated in the hierarchical framework each for area of residence, wealth,

or education, within regions, and in a time-series manner that would capture changes over time. Final estimates of each family planning indicator are the means that were derived from samples of the posterior distribution via the Markov Chain Monte Carlo (MCMC) algorithm. The 95% Credible Intervals (CrI) were calculated as the 2.5 and 97.5 percentiles of the MCMC samples. Model checks included trace plots for convergence of MCMC and Gelman-Rubin diagnostic statistics [35]. 10,000 iterations with three chains, 10 thinning, and 3000 sample burn-in were used in the MCMC algorithm. A dual approach was used to examine the sensitivity of the results. First, by the exclusion of region-level covariate (i.e., HDI), and second, by the alteration of priors for the hyperparameters. A breakdown of the sensitivity analysis is presented as supplementary information (S1 Appendix, S-methods 2). The slope index of inequality (SII) was employed to assess the magnitude of wealth- and education-based disparities for all three indicators for the years 2015 and 2030. The SII shows the absolute difference between the most-advantaged and most-disadvantaged socioeconomic categories in terms of estimated favourable family planning indicators, and the reverse for the adverse indicator (i.e., unmet need). SIIs range between −100 to +100 where 0 indicates no difference [36].

For the determinants analysis, Bayesian hierarchical logistic regression models with random intercepts were utilized. The multilevel framework accounts for dependency across the multilevel data, i.e., individuals nested within households and households within communities. The final model selection was based on comparisons of Deviance Information Criteria (DIC) to assess relative goodness-of-fit. Further details are presented in the supplemental file (S1 Appendix, S-methods 3). All statistical analyses were conducted using STATA version 17 (Stata Corp, College Station, TX, USA).

## Results

### Coverage/disparities by area of residence

Fig 1 illustrates that urban and rural levels of modern contraceptive prevalence and demand satisfied with modern methods steadily increased among married women of reproductive age in Cameroon from 1990 onwards. However, the urban coverage of both indicators has been notably higher compared to the rural coverage. Urban levels of modern contraceptive prevalence were estimated at 29.6% (95% CrI13.0 to 54.4) compared to rural levels of 12.6% (4.9 to 28.8) in

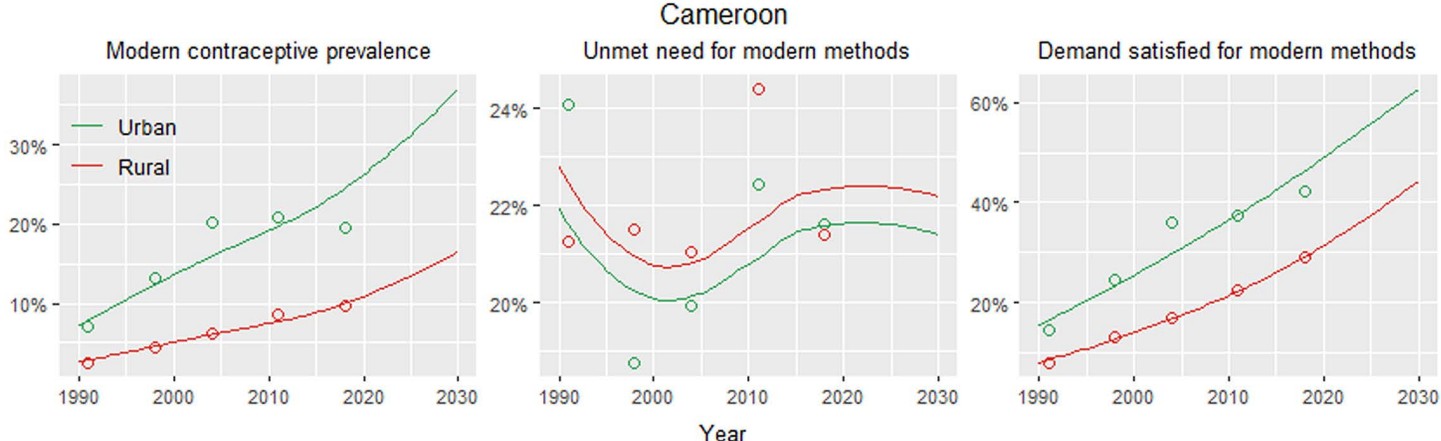

**Fig 1. Urban and rural estimates of use of, unmet need, and demand satisfied for modern contraceptive methods in Cameroon, 1990–2030.** Lines=Loess fit through median estimates; Circles=Demographic and Health Survey estimates.

2023. Even though, the gap could further widen as projections show that urban and rural coverage could increase to 36.9% (17.1 to 61.7) and 16.4% (6.5 to 35.6), respectively, in 2030 (Table 1). As for the unmet need for modern methods, there are no clear patterns of changes over time, and urban levels appear to be only slightly lower than the rural. The unmet need for modern methods in urban and rural areas was estimated at 21.4% (13.6 to 31.9) and 22.2% (14.0 to 33.0) in 2023 and projected to 21.7% (13.7 to 32.5) and 22.4% (14.3 to 33.4) in 2030, respectively.

At the region level, the *Centre* recorded the highest of both urban and rural modern contraceptive prevalence of 49.0% (24.9 to 73.8) and 28.2% (12.4 to 52.3), respectively, in 2023. For the urban areas, the *Centre* is followed by the *Northwest* and *East* with modern contraceptive prevalence of 60.7% (35.9 to 81.9), 49.7% (26.1 to 73.7), and 47.5% (24.3 to 72.2), respectively, projected for 2030 (Table 1). The least estimates seen in the rural areas of the *Adamawa*, *Far North*, and *North* should be sustained based on their respective 2030 projections of 6.9% (2.5 to 17.0), 7.6% (2.8 to 19.0), and 9.5% (3.5 to 23.2). All three northern (i.e., *North*, *Far North*, and *Adamawa*) are among the top five with the widest urban-rural gaps in modern contraceptive prevalence. The two others include the *East* and *Centre*. The *North* and *Far North* recorded the most widening gaps, with respective urban versus (vs) rural modern contraceptive prevalence projections of 56.6% (31.7 to 78.7) vs 9.5% (3.5 to 23.2) and 49.2% (25.7 to 73.2) vs 7.6% (2.8 to 19.0) for 2030. This is compared to the estimated 4.4% (1.6 to 11.7) vs 0.4% (0.1 to 1.1) and 5.1% (1.9 to 12.9) vs 0.5% (0.2 to 1.2), respectively, in 2000. The supplementary material (S3 Table) includes urban- and rural estimates of demand satisfied and unmet need for modern methods.

**Table 1. Urban and rural estimates of modern contraceptive use for Cameroon and its regions, 2000, 2015, and 2030.**

| COUNTRY Region | Urban(U)/ Rural(R) | Modern contraceptive prevalence rate | | |
|---|---|---|---|---|
| | | **2000** | **2015** | **2030** |
| **CAMEROON** | U | 13.8 (5.5–30.8) | 21.5 (9.2–43.8) | 36.9 (17.1–61.7) |
| | R | 5.1 (1.9–13.2) | 8.7 (3.2–21.0) | 16.4 (6.5–35.6) |
| Adamawa | U | 8.2 (3.1–20.1) | 13.8 (5.4–31.5) | 27.4 (11.7–51.5) |
| | R | 1.8 (0.6–4.8) | 3.1 (1.1–8.3) | 6.9 (2.5–17.0) |
| Centre | U | 19.1 (7.8–40.0) | 32.6 (14.7–58.1) | 60.7 (35.9–81.9) |
| | R | 8.7 (3.3–21.6) | 16.4 (6.7–35.9) | 38.3 (18.2–63.3) |
| East | U | 15.0 (6.1–33.1) | 25.5 (10.9–49.4) | 47.5 (24.3–72.2) |
| | R | 5.4 (2.0–13.7) | 9.9 (3.7–23.6) | 22.5 (9.4–44.8) |
| Far North | U | 5.1 (1.9–12.9) | 17.3 (6.8–37.4) | 49.2 (25.7–73.2) |
| | R | 0.5 (0.2–1.2) | 1.7 (0.6–4.9) | 7.6 (2.8–19.0) |
| Littoral | U | 16.1 (6.5–35.6) | 19.1 (7.7–39.1) | 27.3 (11.7–51.3) |
| | R | 11.0 (4.3–25.2) | 13.3 (5.2–29.5) | 19.4 (7.9–40.5) |
| Northwest | U | 14.0 (5.5–31.2) | 26.9 (11.4–51.0) | 49.7 (26.1–73.7) |
| | R | 8.8 (3.3–21.3) | 17.9 (7.5–38.1) | 36.9 (17.3–62.0) |
| North | U | 4.4 (1.6–11.7) | 17.7 (7.0–38.3) | 56.6 (31.7–78.7) |
| | R | 0.4 (0.1–1.1) | 1.7 (0.6–4.8) | 9.5 (3.5–23.2) |
| West | U | 13.8 (5.3–30.9) | 19.3 (7.7–39.8) | 30.3 (13.2–55.0) |
| | R | 9.6 (3.7–23.0) | 13.7 (5.4–30.8) | 22.3 (8.9–45.0) |
| South | U | 19.6 (7.9–40.4) | 21.1 (8.6–43.4) | 29.8 (12.8–54.1) |
| | R | 11.6 (4.5–26.4) | 12.7 (4.9–29.3) | 18.7 (7.2–38.5) |
| Southwest | U | 15.4 (5.9–33.6) | 22.1 (9.0–44.3) | 32.6 (14.7–57.9) |
| | R | 11.2 (4.3–26.2) | 16.5 (6.5–35.5) | 25.2 (10.7–48.6) |

## Coverage/disparities by wealth status

Fig 2 shows trends in demand satisfied with modern methods per wealth quintile and region of Cameroon. Across the board, levels have remained higher among the richer. In 2023, the highest estimate of 82.9% (58.1 to 94.4) demand satisfied with modern methods was estimated among those in the richest quintile of the *Adamawa* region. This was followed by the

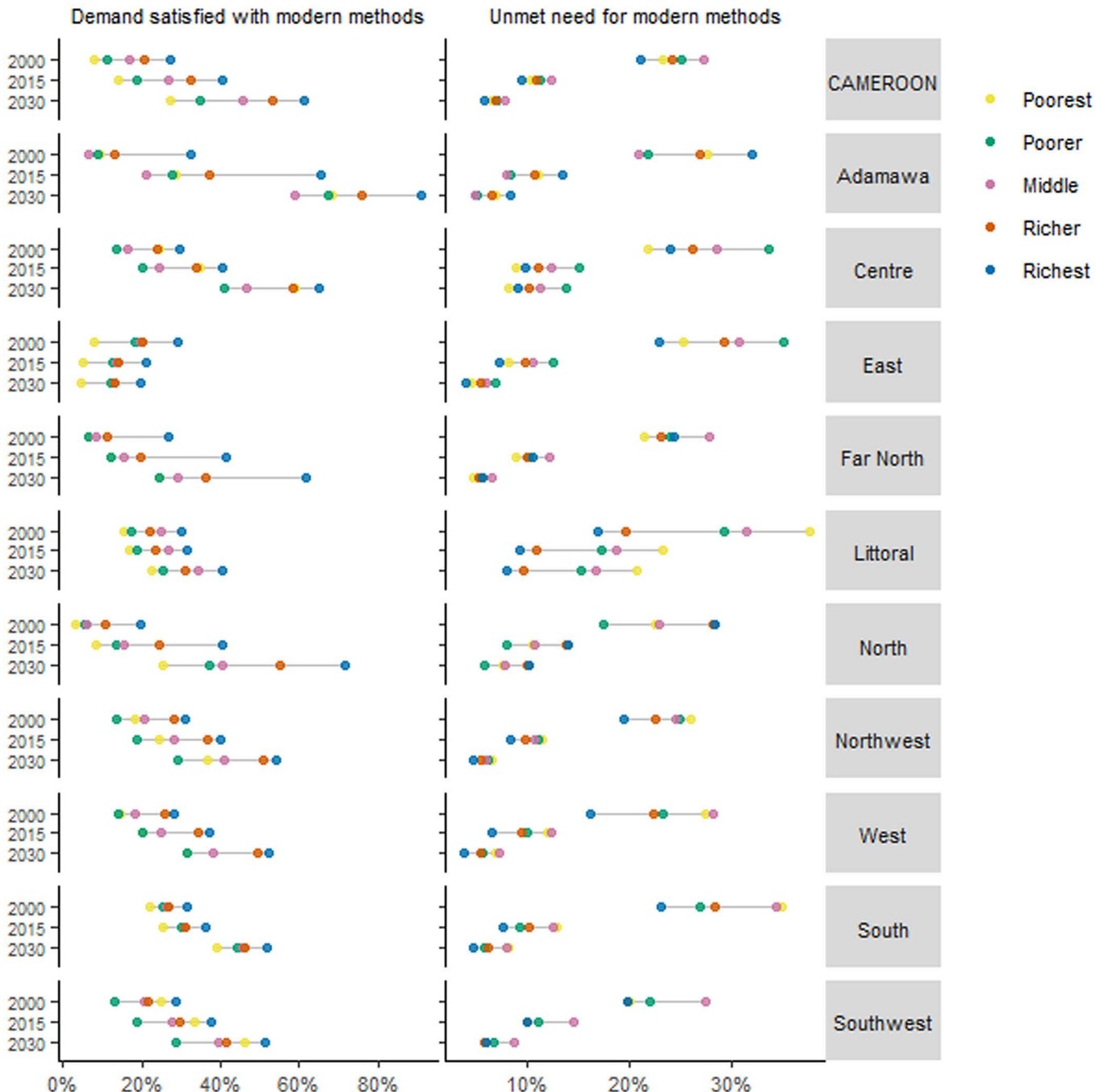

**Fig 2. Demand satisfied and unmet need for modern contraceptive methods by wealth quintile for Cameroon and its regions, 2000, 2015, and 2030.**

richer quintile of *Adamawa* [60.2% (30.2 to 84.0)], richest quintile of the *North* [59.4% (30.1 to 83.9)], and richest quintile of the *Centre* [55.9% (26.9 to 81.3)]. Overall, levels of demand satisfied with modern methods were lower in the poorer wealth quintiles. However, the least estimates included four wealth quintiles in the *East* region, i.e., poorest, poorer, middle, and richer quintile, with 5.3% (1.5 to 16.5), 12.9% (4.1 to 34.5), 14.1% (4.5 to 36.5), and 14.5% (4.6 to 36.4), respectively, in 2023. Notably, only in the *East* were levels shown to decrease over time from 2000 to 2030 across all wealth quintiles. Projections show that the aforementioned top and bottom rankings could be maintained up to 2030. See Fig 2 and the supplemental file (S4 Table). Wealth-based disparities in demand satisfied with modern methods are presented in Table 2. The *Southwest* recorded the narrowest wealth-based gaps, i.e., 9.4 percentage points (%p) (%p) (−12.3 to 31.1) in 2015 and 11.8%p (−14.0–37.6) projected for 2030. This is closely followed by estimates for the *South* and *Centre*. Even though the *Adamawa* shows the widest gaps, it could also record the largest projected decrease, i.e., from 40.7%p (10.2 to 71.2) in 2015 to 27.0%p (2.3 to 51.6) in 2030.

Fig 2 also includes unmet need for modern methods estimates between 2000 and 2030 across wealth quintiles and regions of Cameroon. Negative trends are observable for each of the quintiles across all regions, and most substantially between 2000 and 2015. Similar to demand satisfied, unmet need for modern methods tends to be more favourable among the richer quintiles. Wherefore, richer quintiles registered lower while poorer quintile showed higher levels of unmet need for modern methods. In 2023, the lowest estimate was registered among the richest in the *West* region [5.1% (1.8 to 13.5)], where according to projections could reach as low as 3.7% (1.3 to 10.0) in 2030. Current (2023) levels are also lower among the richest quintiles in the *East*, *Northwest*, and *South*, poorer and middle quintiles in the *Adamawa*, and poorest quintiles in the *East* and *Far North*. Uniquely, the poorer groups in the *Far North* and *Centre* regions showed lower prevalence of unmet need for modern methods compared to the wealthier. The other end with the highest estimates includes three out of the five quintiles within the *Littoral* region, i.e., the poorest, middle, and poorer quintiles with 23.1% (9.4 to 47.4), 18.6% (7.3 to 40.1), and 17.2% (6.9 to 37.3), respectively, in 2023. Despite the decreases, levels for these three quintiles are to remain the highest with forecasts of at least

**Table 2. Magnitude of socio-economic inequalities in demand satisfied for modern methods for Cameroon and its regions, 2015 and 2030.**

| COUNTRY Region | Slope Index of Inequality (SII) for demand satisfied with modern methods (%p) | | | |
|---|---|---|---|---|
| | Wealth-based SII | | Education-based SII | |
| | 2015 | 2030 | 2015 | 2030 |
| CAMEROON | 33.1 (31.8–34.5) | 41.9 (40.4–43.3) | 56.0 (44.9–67.0) | 61.0 (49.2–72.8) |
| Adamawa | 40.7 (10.2–71.2) | 27.0 (2.3–51.6) | 78.9 (73.7–84.0) | 79.3 (73.9–84.7) |
| Centre | 12.6 (-10.6–35.8) | 14.9 (−13.6–43.4) | 39.1 (29.3–48.8) | 34.7 (24.8–44.5) |
| East | 16.9 (10.0–23.8) | 15.9 (9.5–22.2) | 39.6 (31.1–48.0) | 45.0 (33.9–56.0) |
| Far North | 33.4 (16.1–50.7) | 42.3 (21.0–63.7) | 33.5 (−15.6–82.5) | 47.6 (−7.9–99.7) |
| Littoral | 17.3 (13.3–21.3) | 20.5 (15.6–25.4) | 31.6 (15.6–47.6) | 33.5 (16.2–50.9) |
| Northwest | 24.2 (12.9–35.4) | 27.9 (13.5–42.4) | 24.6 (19.5–29.7) | 29.0 (23.5–34.5) |
| North | 37.9 (30.9–44.9) | 53.1 (45.8–60.4) | 59.2 (32.8–85.5) | 70.4 (51.2–89.6) |
| West | 24.0 (18.5–29.5) | 29.1 (22.1–36.2) | 44.6 (39.5–49.6) | 47.8 (41.7–54.0) |
| South | 11.4 (7.8–15.1) | 13.6 (9.7–17.5) | 35.6 (26.9–44.4) | 40.1 (30.5–49.8) |
| Southwest | 9.4 (−12.3–31.1) | 11.8 (−14.0–37.6) | −10.0 (−53.5–33.5) | −6.8 (−41.5–28.0) |

%p= percentage points.

15.2% (5.9 to 33.8) (among the poorer) for 2030. Wealth-based estimated values and disparities for all three family planning indicators can be found in supplemental files S4 and S5 Tables, respectively.

## Coverage/disparities by education level

Estimates of demand satisfied with and unmet need for modern methods by level of education are presented in Fig 3. For demand satisfied with modern methods, except for the *South* region, levels are shown to steadily increase across all educational categories from 2000 to 2030. Like with the wealth-based assessment, those with higher educational attainment tended to record higher levels of demand satisfied with modern methods and vice versa. Levels among those with higher education in *Adamawa* were highest at 85.9% (69.5 to 94.2) in 2023 and 89.5% (76.1 to 95.8) projected for 2030. Oddly, this was followed by those with no education in the *Southwest* [75.0% (52.2 to 89.2) in 2023] with levels that are higher than in the other groups with higher educational attainment within the same region. Ranked third and fourth are those with higher education in the *Centre* and secondary education in the *North*, respectively. At the bottom with the lowest demand satisfied with modern methods of 8.6% (3.3 to 20.4) in 2023 was the education category 'none' of the *Far North* amongst whom levels are to remain lowest up to 2030 [11.8% (4.7 to 26.5)]. 'None' education groups in the *South* and *Adamawa* closely follow. The education-based disparities in demand satisfied with modern methods (Table 2) were narrowest but negative in the *Southwest* region which has a 2030 projection of -6.8%p (−41.5–28.0). The largest gaps were seen in the *Adamawa* and *North* regions where the respective magnitudes are estimated to further widen to 79.3%p (73.9 to 84.7) and 70.4%p (51.2 to 89.6) in 2030.

Regarding the unmet need for modern methods, only the *Southwest* and *East* show decreases across all four education levels from 2000 to 2030 (Fig 3). Amidst falling levels, these two regions have the most favourable projections in terms of low unmet need for modern methods, particularly among the 'none' and 'higher' educated, respectively. Lower levels are also projected across all four educational categories in both the *Southwest* and *East* especially compared to the other regions. As of 2023, the least prevalence was among the higher educated in the *North* at 6.4% (3.1 to 12.8), even though levels could slightly rise to 7.3% (3.5 to 14.7) in 2030. At the other end, those with higher education in *Adamawa* showed a noticeably higher unmet need for modern methods, compared to the region's other (lower) education categories as well as overall. Prevalence in this group reached 54.1% (35.3 to 71.7) in 2023 and is projected to slightly increase to 55.0% (36.2 to 72.7) in 2030. Projections are least favourable in the *Far North* due to recording the steepest increases across all educational groups. Supplemental files S5 and S6 Tables respectively show disparities and prevalence for each family planning indicator per level of education.

## Determinants of family planning

The descriptive results (Table 3) indicate that among the 1,513 respondents, the average age was 30.8 years (SD ± 7.9), with a considerable proportion falling within the 20–29 (38.6%) and 30–39 (37.9%) age groups, while only 7.7% were aged 15–19. The majority were HIV-negative (95.8%), with only 4.2% testing positive. Most participants had secondary education (43.8%), followed by primary (32.6%). The religious affiliation was predominantly Catholic (38.4%) and Christian (37.6%). Half of the participants (50.4%) had 1–3 living children, and a notable portion (31.0%) reported no media exposure. The wealth distribution was relatively balanced across the poorer to richest quintiles (between 20.4 and 25.3%), while the poorest quintile

made up 12.7%. Participants were nearly evenly split between rural (48.1%) and urban (51.9%) residences, with the *Centre* region having the highest representation (21.0%).

In Table 3, adjusted odds ratios (AORs) from the determinants analysis of modern contraceptive use and unmet need for modern methods among married women of reproductive age in Cameroon are presented. Individual level variables, age ≥ 20 years, higher education level, practising Catholic or Christian religion, having one or more living child(ren), and higher

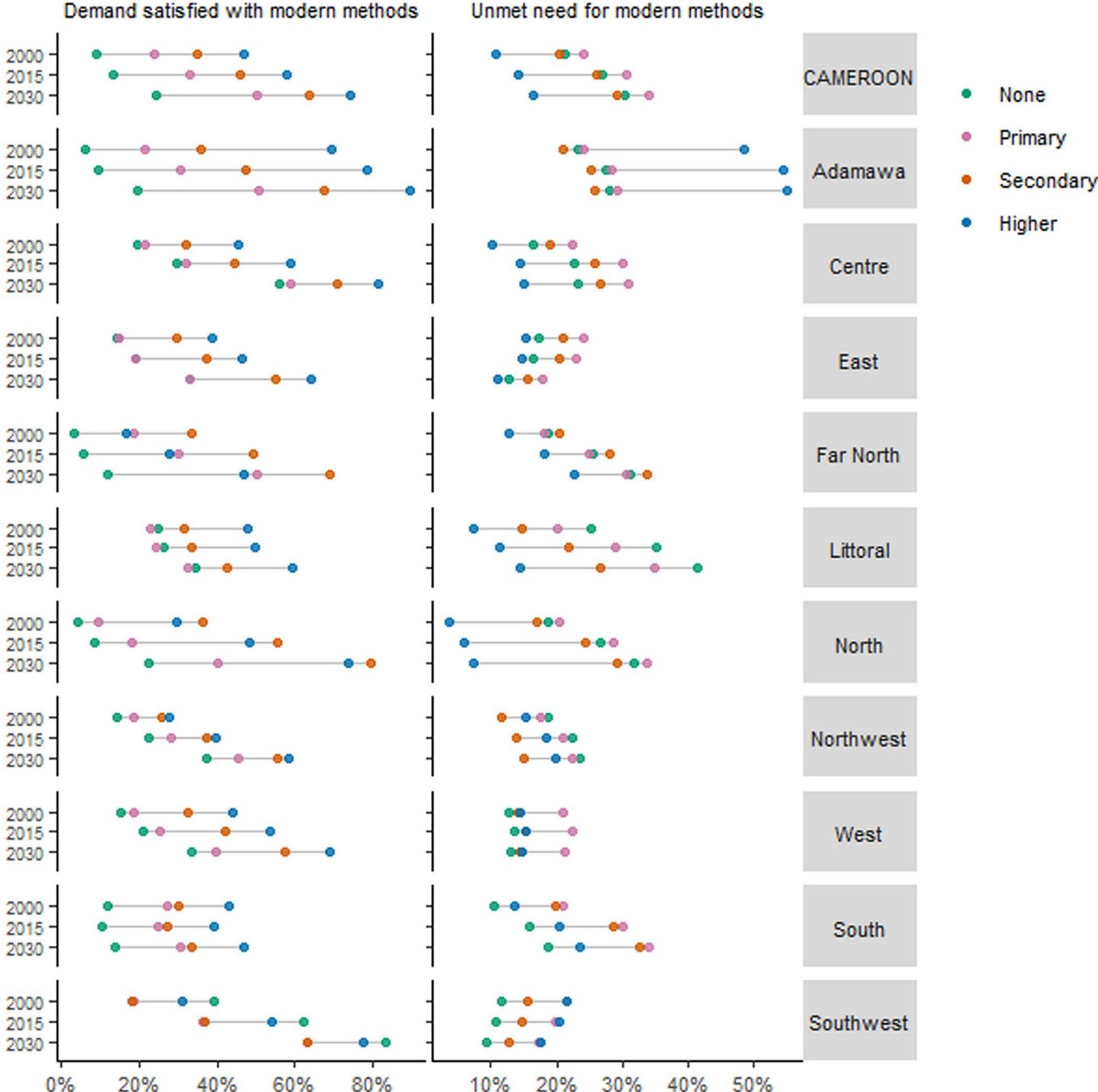

**Fig 3. Demand satisfied and unmet need for modern contraceptive methods by level of education for Cameroon and its regions, 2000, 2015, and 2030.**

**Table 3. Determinants of use of and unmet need for modern methods of family planning in Cameroon, DHS 2018.**

| Variable | Categories | Adjusted Odds Ratio (95% Credible Interval) | | N (%) |
|---|---|---|---|---|
| | | Modern contraceptive use | Unmet need for modern methods | |
| HIV status | Negative | 1 | 1 | 1,450 (95.8) |
| | Positive | 0.85 (0.68, 1.03) | 1.10 (0.88, 1.38) | 63 (4.2) |
| Age group, years | 15–19 | 1 | 1 | 117 (7.7) |
| | 20–29 | 2.23 (1.96, 2.62) | 0.42 (0.35, 0.51) | 584 (38.6) |
| | 30–39 | 2.20 (1.84, 2.61) | 0.40 (0.33, 0.47) | 573 (37.9) |
| | 40–49 | 1.61 (1.32, 1.95) | 0.52 (0.41, 0.67) | 239 (15.8) |
| Level of education | None | 1 | 1 | 253 (16.7) |
| | Primary | 2.56 (2.08, 3.11) | 0.49 (0.38, 0.61) | 493 (32.6) |
| | Secondary | 3.11 (2.74, 3.64) | 0.35 (0.28, 0.42) | 662 (43.8) |
| | Higher | 2.43 (1.97, 3.00) | 0.33 (0.26, 0.40) | 105 (6.9) |
| Religion | Muslim | 1 | 1 | 317 (21.0) |
| | Catholic | 1.38 (1.15, 1.70) | 0.77 (0.66, 0.90) | 581 (38.4) |
| | Christian | 1.35 (1.15, 1.58) | 0.64 (0.57, 0.73) | 569 (37.6) |
| | None/other | 0.37 (0.30, 0.45) | 1.67 (1.35, 2.09) | 46 (3.0) |
| Living children | None | 1 | 1 | 37 (2.4) |
| | 1–3 | 1.32 (1.16, 1.49) | 0.97 (0.78, 1.19) | 762 (50.4) |
| | 4–6 | 1.73 (1.39, 2.14) | 0.83 (0.72, 0.97) | 547 (36.2) |
| | ≥6 | 1.48 (1.19, 1.85) | 1.43 (1.12, 1.79) | 167 (11.0) |
| Media exposure | None | 1 | 1 | 469 (31.0) |
| | < Once a week | 1.17 (0.96, 1.40) | 0.76 (0.67, 0.87) | 602 (39.8) |
| | > Once a week | 1.14 (0.94, 1.31) | 0.92 (0.76, 1.11) | 440 (29.1) |
| Wealth quintile | Poorest | 1 | 1 | 192 (12.7) |
| | Poorer | 2.03 (1.83, 2.24) | 0.59 (0.46, 0.76) | 308 (20.4) |
| | Middle | 2.27 (1.91, 2.67) | 0.45 (0.37, 0.55) | 383 (25.3) |
| | Richer | 2.07 (1.64, 2.63) | 0.48 (0.39, 0.58) | 309 (20.4) |
| | Richest | 3.04 (2.42, 3.74) | 0.34 (0.29, 0.42) | 321 (21.2) |
| Residence | Rural | 1 | 1 | 727 (48.1) |
| | Urban | 1.01 (0.82, 1.23) | 0.91 (0.77, 1.09) | 786 (51.9) |
| Region | Adamawa | 1 | 1 | 95 (6.3) |
| | Centre | 1.51 (1.28, 1.76) | 0.70 (0.59, 0.82) | 318 (21.0) |
| | East | 2.34 (1.85, 2.89) | 0.13 (0.10, 0.15) | 166 (11.0) |
| | Far North | 2.18 (1.72, 2.70) | 0.49 (0.41, 0.59) | 132 (8.7) |
| | Littoral | 1.18 (1.02, 1.35) | 0.81 (0.67, 0.97) | 213 (14.1) |
| | Northwest | 1.92 (1.55, 2.37) | 0.60 (0.47, 0.77) | 138 (9.1) |
| | North | 1.95 (1.64, 2.34) | 0.45 (0.34, 0.60) | 87 (5.8) |
| | West | 1.49 (1.29, 1.78) | 0.37 (0.30, 0.45) | 171 (11.3) |
| | South | 0.93 (0.73, 1.18) | 1.56 (1.28, 1.91) | 162 (10.7) |
| | Southwest | 1.60 (1.18, 2.18) | 0.99 (0.86, 1.18) | 31 (2.0) |

Modern methods=Modern contraceptive methods; Media exposure represents a combination of variables for frequency of reading newspapers or magazines, listening to radio, watching television, and use of internet.

household wealth quintile, were associated with increased odds of modern contraceptive use, credible intervals inclusive. The odds were higher by 2.23 (1.96 to 2.62) among those aged 20 to 29 years compared to the 15- to 19-year-olds; 2.43 (1.97 to 3.00) among the 'higher' compared to the 'none' educated; 1.38 (1.15 to 1.70) among Catholics relative Muslims; 1.73 (1.39

to 2.14) for those with 4 to 6 relative to no living children; and 3.04 (2.42 to 3.74) among those that lived in households ranked in the richest compared to the poorest quintile. Except in the *South*, living in every other region showed higher odds of modern contraceptive use relative to the *Adamawa*. AORs for unmet need for modern methods mostly reflect the converse of AORs for modern contraceptive use. Crude odds ratios are presented in the supplemental file S7 Table.

## Sensitivity analysis

The overall absolute differences between the main versus altered models for each family planning indicator and socioeconomic category in 2000, 2015, and 2030 were very minor. Excluding region-level predictors, the differences for use of, unmet need for, and demand satisfied with modern methods respectively ranged between: 0.01 to 0.08%, 0.01 to 0.04%, and 0.00 to 0.07% by area of residence (Supplemental file, S8 Table); 0.00 to 0.03%, 0.00 to 0.05%, and 0.01 to 0.02% by wealth quintile (Supplemental file, S9 Table); and 0.01 to 0.26%, 0.02 to 0.21, and 0.00 to 0.26% by education level (Supplemental file, S10 Table). Altering priors for the hyperparameters, the differences for use of, unmet need for, and demand satisfied with modern methods respectively ranged between: 0.01 to 0.04%, 0.00 to 0.02%, and 0.02 to 0.06% by area of residence (Supplemental file, S11 Table); 0.00 to 0.14%, 0.00 to 1.76%, and 0.00 to 0.20% by wealth quintile (Supplemental file, S12 Table); and 0.06 to 0.15%, 0.01 to 0.05%, and 0.11 to 0.19% by education level (Supplemental file, S13 Table).

## Discussion

In this paper, area of residence-, wealth-, and education-based levels, projections of, and disparities in family planning among married women of reproductive age across the 10 regions of Cameroon were presented. Results for most of the regions revealed higher levels of favourable indicators, i.e., modern contraceptive prevalence and demand satisfied with modern methods, among those in the higher ranked socioeconomic categories such as urban residence, richest wealth quintile, and higher education. For the unfavourable indicator (unmet need for modern methods), estimates mostly reflected the reverse, i.e., higher levels in the lower-ranked socioeconomic categories like rural residence, poorest wealth quintile, and no education.

Across all 10 regions, the analysis highlights a clear trend: urban areas demonstrate noticeably higher levels of modern contraceptive use in urban compared to rural areas. These findings were not unexpected and they align with disparities recorded from previous surveys conducted at the national level [37]. Studies in other SSA countries [8,13,23] have likewise shown noticeably higher urban compared to rural modern contraceptive prevalence at the national level. Similar to the urban trends, all 10 rural settings recorded substantial increases in modern contraceptive prevalence that are projected to continue up to 2030. The *Centre* region stands out because coverage in its rural area is noticeably higher than in the other rural and in 6 out of the 10 urban areas. This wide disparity can be attributed to better access to family planning services, as the *Centre* region is the administrative capital and has a higher concentration of health resources, including per capita trained personnel, facilities, and health budget, compared to other regions [16,38]. Additionally, HIV rates may play a significant role in this context, particularly due to their influence on the use of barrier contraceptives such as male and female condoms [25–27]. Recent data indicate that certain rural divisions within the Centre region, including *Haute Sanaga*, *Nyong et Mfoumou*, *Nyong et So'o*, and *Haut Nyong*, have emerged as major hot-spots of HIV infection in Cameroon [39]. This correlation suggests that higher HIV prevalence could drive increased reliance on condom use for both prevention and family

planning purposes. On the other end, rural areas in the northern regions (*Adamawa*, *Far North*, and *North*) showed substantially lower modern contraceptive prevalence compared to other rural and urban areas. It is worth noting that these three regions each receive much less in terms of health resources per capita compared to the other 7 regions [16,38]. In the *Far North*, for instance, the per capita health budget and facilities/personnel are, respectively, only one-third and less than half that of the *Centre* [16]. Additionally, given that illiteracy rates are also highest in the aforementioned northern regions (at least 60% each) [40], lack of or limited access to information could be an important contributor, particularly in rural areas. An important policy focus should thus be to target disparities by area of residence, especially in regions such as the *Far North* and *North* where urban-rural gaps are widening and projected to persist long term.

Previous national-level assessments showed much larger wealth compared to education-related disparities in the demand satisfied with modern methods [6]. However, the regional-level analysis conducted herewith uncovered much wider education- compared to wealth-based gaps, particularly in the northern regions of *Adamawa*, *Far North*, and *North*. Wider disparities across the cited settings may be linked to underlying socioeconomic disadvantages compared to the other regions of the country. For instance, the Human Development Index (a composite measure that incorporates life expectancy, education, and per capita income [32]) for each of the northern regions is less than that for the other regions as well as the national average [31]. In a previous study, the same three regions were found to have the lowest levels of demand satisfied with modern methods in Cameroon [10]. Having shown the widest wealth- and education-based gaps, despite also recording the highest estimates among the richer and more educated groups (particularly the *Adamawa* and *North*), this indicates that demand satisfied with modern methods in the northern regions is being pulled down by levels among the lower ranked socioeconomic categories. These findings demonstrate the need to prioritise the poorer and less educated during family planning programming. Simultaneously, the country should focus on increasing literacy and reducing poverty through its development initiatives as well as collaborative efforts with social services organisations, to enhance understanding and empower decision-making in family planning, particularly for these disadvantaged groups [41]. Meanwhile, for regions with the smallest disparities in demand satisfied with modern methods, only in the *Southwest* did this cut across both the wealth- and education-based assessments. Features that were earlier linked to performances in the other regions such as the socioeconomics, availability of health resources, or relevant health indicators like HIV rate are not distinguished in this region. The *Southwest* historically stands out as an epicentre of higher learning institutions in Cameroon [42]. Whether this transcends to an environment where the less educated have increased odds of an environment with more access to information on issues like contraceptive use is unclear. Nonetheless, community-level assessments are recommended in better-performing regions like *Southwest* to understand policy-applicable elements that can be replicated elsewhere to reduce disparities and enhance coverage of demand for family planning satisfied among socioeconomically disadvantaged populations.

Regarding the unmet need for modern methods, wealth categories across all 10 regions showed positive progress in terms of declining levels and narrowing disparities over time. Notably, the findings reflect more improvements in the lower-ranked wealth quintiles, as evidenced by the negative SIIs (Supplemental file, S5 Table). This trend indicates that proportions of married women of reproductive age whose need for birth spacing or limiting is being met using modern contraceptive methods in each region are increasingly from poorer households. More unclear are the education-related patterns of unmet need for modern methods, as

trends across categories showed increasing levels in most regions (6 out of 10), and decreasing or steady in the rest. Similarly, irregular education-based patterns of unmet need have been reported across regions of Ghana [20]. It is noteworthy that rising levels might not necessarily indicate a regress in performance as trajectories of unmet need are said to largely depend on the contraceptive prevalence level [43]. A multi-country assessment demonstrated that levels of unmet need tend to initially rise until contraceptive prevalence reaches around 20% where an accelerated decrease sets in [43]. Notably, more than half of the education categories across the 10 regions recorded an average modern contraceptive prevalence of less than 20% over the study period. Careful consideration of contraceptive prevalence is therefore required during family planning strategy development where unmet need is among key indicators of performance.

The determinants analysis is relevant in terms of developing comprehensive initiatives aimed at promoting the use of modern contraceptives for family planning. Age, education, wealth, having living children, and religion were found to be significant predictors of either the use or nonuse of modern contraceptives. Overall, the findings point toward the need for family planning interventions to prioritise married women of reproductive age who are of teenage age, not/less educated, living in the poorest households, have no living children, practice Islam, and/or reside in regions such as *Adamawa*. Estimates showing that women aged 20 to 29 and 30 to 39 are at least twice more likely to use modern contraceptives compared to 15 to 19-year-olds aligns with literature, which indicates that contraceptive use trends increase with age until they begin to decline in older generations [23,44]. This finding highlights the importance of targeting younger women in family planning programmes. The practice of Catholicism/Christianity being linked to higher odds of modern contraceptive use compared to Islam, reinforces previous findings that religious beliefs significantly influence contraceptive choices [17]. Higher odds of modern contraceptive use with higher parity (number of living children) underscores the role of childbearing experience in family planning [17–19]. Notably, the study shows unexpectedly lower rates of modern contraceptive use among women with higher compared to secondary education, in the richer compared to middle wealth quintile, and parity ≥ 6 compared to parity 4–6. Recognising the potential influence of study limitations, these findings suggest that women with higher educational attainment or from wealthier households may prioritise career or personal development over immediate reproductive goals. Also, women with more children may experience "fertility inertia" (be less likely to seek contraception after achieving their desired family size). The stark contrast in modern contraceptive use based on level of education and household wealth aligns with findings from the trend analyses, emphasizing the need for targeted interventions to improve family planning coverage in lower socioeconomic strata. Geographic disparities in modern contraceptive use also indicate the necessity for region-specific strategies. Our findings align with prior assessments that failed to incorporate HIV status among covariates [17–23]. Nevertheless, incorporating HIV status into future studies could provide deeper insights into contraceptive behaviours, since higher rates of modern contraceptive (particularly condom) use have been reported among HIV-positive women [45,46]. Overall, targeted contraception expansion efforts are crucial in Cameroon as those in the lower socioeconomic ranks would benefit more [47] from improved availability, accessibility, or affordability.

This study not only contributes in terms of uncovering disparities to inform family planning policy formulation but could benefit local-level monitoring of progress toward global benchmarks such as the proposed SDG for attainment of ≥ 75% demand satisfied with modern methods by 2030 [6]. The findings presented should enable Cameroon to adopt varied approaches for family planning implementation and expansion, tailored to the disparities

identified in each region. However, it is not without limitations. Firstly, this relates to the uncertainty associated with data for smaller areas (i.e., socioeconomic categories per region) as Cameroon-DHS data is primarily powered at the region level [37]. Extra levels of hierarchy representing the socioeconomic categories, regions, and time were thus introduced to allow for greater pooling of information across the smaller populations [48]. However, the findings should still be interpreted with caution. Furthermore, due to other data limitations, indirect impacts of important processes such as family planning policy changes were not factored into the model. While such processes could be reflected in incorporated variables like the HDIs and/or captured in the data collected at each survey point, we aimed to generate estimates mainly based on current trends. Regarding the determinants analysis, the causal links between examined factors relating to the use- and non-use of modern contraceptives are difficult to establish due to the study's cross-sectional nature. Additionally, this study excludes unmarried women of reproductive age even though they make up increasing proportions of users of family planning services [49]. Given the discrepancies in definitions of the demand for family planning between the two groups [29,49], we aim to conduct a separate assessment for the unmarried population.

## Conclusion

Wide urban-rural, wealth-, and education-based disparities remain in the coverage of modern contraceptive methods of family planning among married women of reproductive age across the regions of Cameroon. At most, ≥ 75% demand satisfied with modern methods could be attained in only one-third of rural areas, one-sixth of the two poorest quintiles, and one-quarter of two bottom educational categories by 2030. These findings underscore the urgent need for region-level family planning policies that prioritise specific groups, particularly rural, less educated, and impoverished women. Additionally, the determinants analysis emphasizes the importance of factoring age group, number of living children, level of education, wealth status, and religion into the design of family planning interventions. To effectively address these disparities, multi-sectoral cooperation is essential. This includes initiatives aimed at improving literacy rates, implementing social support schemes, providing subsidised services, and enhancing access to family planning resources. Future studies examining locality-specific influences on the uptake of family planning services and commodities will be crucial to further inform programmes per region.

## Supporting information

**S1 Table. Characteristics of data used to assess family planning indicators by area of residence, wealth, and educational status across regions of Cameroon.** DHS = Demographic and Health Survey; mCPR = modern contraceptive prevalence; mUNFP = unmet need for modern methods; mDSFP = demand satisfied with modern methods. Sample = women aged 15 to 49 years who are married or in a union.
(DOCX)

**S2 Table. Definitions of the coverage for family planning indicators.** Married women include those who are married or in a union. Total number of women with a demand for family planning (modern contraceptive prevalence + traditional contraceptive prevalence + unmet need). Modern contraceptive methods include sterilisations, oral contraceptive pills, intrauterine devices, injectables, implants, condoms, lactational amenorrhea method, standard days method, emergency contraception and vaginal barrier methods.
(DOCX)

**S3 Table. Use of, unmet need, and demand satisfied for modern contraceptive methods by rural and urban residence across regions of Cameroon, 2015 and 2030.** Estimates are in % (95% Credible Interval)
(DOCX)

**S4 Table. Use of, unmet need, and demand satisfied for modern contraceptive methods by wealth quintile across regions of Cameroon, 2015 and 2030.** Estimates are in % (95% Credible Interval); Q1 = poorest, Q2 = poorer, Q3 = middle, Q4 = richer, Q5 = richest.
(DOCX)

**S5 Table. Magnitude of socio-economic inequalities in the use of, unmet need, and demand satisfied for modern contraceptive methods across regions of Cameroon, 2015 and 2030.** %p = percentage points.
(DOCX)

**S6 Table. Use of, unmet need, and demand satisfied for modern contraceptive methods by level of education across regions of Cameroon, 2015 and 2030.** Estimates are in % (95% Credible Interval); E1 = none, E2 = primary, E3 = secondary, E4 = higher levels of educational attainment.
(DOCX)

**S7 Table. Determinants of use of and unmet need for modern methods of family planning in Cameroon, DHS 2018.** Modern methods=Modern contraceptive methods; Media exposure represents a combination of variables for frequency of reading newspapers or magazines, listening to radio, watching television, and use of internet; CrI = Credible Interval.
(DOCX)

**S8 Table. Differences in posterior means of main and altered (no predictor) models by area of residence.** HDI = Human Development Index.
(DOCX)

**S9 Table. Differences in posterior means of main and altered (weak priors) models by area of residence.**
(DOCX)

**S10 Table. Differences in posterior means of main and altered (no predictor) models by wealth quintile.** HDI=Human Development Index; Q1 = poorest, Q2 = poorer, Q3 = middle, Q4 = richer, and Q5 = richest.
(DOCX)

**S11 Table. Differences in posterior means of main and altered (weak priors) models by wealth quintile.** Q1 = poorest, Q2 = poorer, Q3 = middle, Q4 = richer, and Q5 = richest.
(DOCX)

**S12 Table. Differences in posterior means of main and altered (no predictor) models by education level.** HDI=Human Development Index; E1 = none, E2 = primary, E3 = secondary, E4 = higher.
(DOCX)

**S13 Table. Differences in posterior means of main and altered (weak priors) models by education level.** E1 = none, E2 = primary, E3 = secondary, E4 = higher.
(DOCX)

**S1 Appendix. S-Methods: Bayesian analysis.**
(DOCX)

## Author contributions

**Conceptualization:** Raïssa Shiyghan Nsashiyi, Md Mizanur Rahman, Lawrence Monah Ndam, Masahiro Hashizume.

**Formal analysis:** Raïssa Shiyghan Nsashiyi, Lawrence Monah Ndam.

**Methodology:** Raïssa Shiyghan Nsashiyi, Md Mizanur Rahman.

**Supervision:** Md Mizanur Rahman, Lawrence Monah Ndam, Masahiro Hashizume.

**Writing – original draft:** Raïssa Shiyghan Nsashiyi.

**Writing – review & editing:** Raïssa Shiyghan Nsashiyi, Md Mizanur Rahman, Lawrence Monah Ndam, Masahiro Hashizume.

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
