## [Decision Letter · Decision Letter 0]

16 Oct 2024

PONE-D-24-06584The Editor-in-Chief, PLOS ONE journal,

A subnational socioeconomic assessment of family planning levels, projections, and disparities among married women of reproductive age in CameroonPLOS ONE

Dear Dr. Nsashiyi,

Thank you for submitting your manuscript to PLOS ONE. After careful consideration, we feel that it has merit but does not fully meet PLOS ONE’s publication criteria as it currently stands. Therefore, we invite you to submit a revised version of the manuscript that addresses the points raised during the review process.

We look forward to receiving your revised manuscript.

Kind regards,

Yibeltal Alemu Bekele, MpH

Academic Editor

PLOS ONE

**Journal Requirements:**

2. Please amend either the title on the online submission form (via Edit Submission) or the title in the manuscript so that they are identical.

Reviewers' comments:

Reviewer's Responses to Questions

**Comments to the Author**

1. Is the manuscript technically sound, and do the data support the conclusions?

Reviewer #1: Yes

Reviewer #2: Yes

2. Has the statistical analysis been performed appropriately and rigorously? 

Reviewer #1: I Don't Know

Reviewer #2: Yes

3. Have the authors made all data underlying the findings in their manuscript fully available?

Reviewer #1: Yes

Reviewer #2: Yes

4. Is the manuscript presented in an intelligible fashion and written in standard English?

Reviewer #1: Yes

Reviewer #2: Yes

5. Review Comments to the Author

**Reviewer #1:**  Please find my comments below:

First of all, it is important to know which definition of unmet need used in the manuscript.

Also, It is important to provide information on average marriage age, especially for women.

In the line 96, the writing of the figure 1,5426 is not correct

In the lines 102 and 103, the method can be used by woman or her husband/ partner

Regarding the line 336 and next sentences It is very important to prioritize the poorer and less educated people. But simultaneously, country should develop programs for increasing the literacy level as a standing agenda and a cornerstone for development issues. The social policy and social security relevant organization need to develop the relevant programs.

In the line 354 what is the meaning of positive trend. Does it mean the gap decrease or the figure increase. For me it is vague.

In the line 366 In the text, more than 20 years old is a predictor for increasing contraceptive use. As a usual, with increasing the age and going to the older generation, contraceptive use may decrease. It is necessary to have a short explanation on it.

In the line 376 and next sentences, uncovering disparities and its reasons will help policymakers to develop a plan toward SDG goals. It will show that different region in the country needs different approaches and intervention.

In the conclusion, it is necessary to think about multi-sectoral cooperation for improvement of literacy rate, social support schemes, provision of services with subsidy, improvement of access ...

**Reviewer #2: ** There is no substantial comment to be forwarded, How ever, I recommend the abstract to be rewritten to give a brief and explained background information regarding the topic "interest socioeconomic assessment of family planning levels". Additionally, the conclusion section should be revised to make it focused on the major results.

6. PLOS authors have the option to publish the peer review history of their article (what does this mean? ). If published, this will include your full peer review and any attached files.

**Do you want your identity to be public for this peer review?** For information about this choice, including consent withdrawal, please see our Privacy Policy .

Reviewer #1: **Yes: ** Mohammad Eslami

Reviewer #2: No

---

## [Author Response · Author response to Decision Letter 0]

28 Oct 2024

Response to Reviewers

Journal Requirements:

Response to point 1: Thank you. The manuscript has been revised in line with the PLOS ONE's style requirements provided in the linked documents.

2. Please amend either the title on the online submission form (via Edit Submission) or the title in the manuscript so that they are identical.

Response to point 2: Thank you. The title on the online submission form has been amended to be identical to the title in the manuscript.

Response to point 3: Thank you. The reference list has been crosschecked and updated according to the specifications mentioned above.

Three references have been added to the reference list to support revisions made, including some in line with some of the reviewers’ comments. These include;

Citation [41] (Lines 356–359)

• Reference: 41. Sultan S. The Effects of Education, Poverty, and Resources on Family Planning in Developing Countries. Clinics in Mother and Child Health. 2018;15. doi: 10.4172/2090-7214.1000289

• Rationale: To support revision made in response to Reviewer #1’s Comment 6 below.

Citation [44] (Lines 394–397)

• Reference: 44. Nimani TD, Tadese ZB, Tadese EE, et al. Trend, geographical distribution, and determinants of modern contraceptive use among married reproductive-age women, based on the 2000, 2005, 2011, and 2016 Ethiopian demographic and health survey. BMC women's health 2023;23(1):629. doi: 10.1186/s12905-023-02789-z

• Rationale: To support revision made in response to Reviewer #1’s Comment 8 below.

Citation [10] (Lines 70–71; 349–352)

• Reference: 10. Nsashiyi RS, Rahman MM, Ndam LM, et al. Contraceptive use, unmet need, and demand satisfied for family planning across Cameroon: a subnational study including indirect effects of COVID-19 and armed conflict on projections. BMC Global and Public Health 2024;2(1):40. doi: 10.1186/s44263-024-00071-4

• Rationale: To insert a more relevant recently published article that aligns with the statements made.

Reviewer #1: Please find my comments below:

Comment 1: First of all, it is important to know which definition of unmet need used in the manuscript.

Response 1: Thank you for your feedback.

We defined “Unmet need for modern methods is the percentage of women who are not currently using any method of contraception to prevent pregnancy but want to space or limit childbearing.” (Lines 105–107)

The following statement has been added for clarity “...This indicator also includes women using traditional family planning methods, as they are considered to have an unmet need for more effective modern contraceptive methods. (Lines 107–109)

Our definition was drawn from the DHS revised definition of unmet need (Bradley et al. (2012) [Citation number 29]: https://dhsprogram.com/pubs/pdf/AS25/AS25[12June2012].pdf), which also includes women who are using a traditional method of family planning.

Comment 2: Also, It is important to provide information on average marriage age, especially for women.

It is important to provide information on average marriage age, especially for women.

Response 2: Thank you. Average marriage age has been added to the descriptive results (i.e., in Line 266–267)

Comment 3: In the line 96, the writing of the figure 1,5426 is not correct

Response 3: Thank you. The figure has been corrected to “15,426” (Line 99)

Comment 4: In the lines 102 and 103, the method can be used by woman or her husband/ partner

Response 4: Thank you. The definition has been revised to include husbands/partners (Line 104).

Comment 5: There are something which not follow the main trend. Their should be considered in the discussion.

There were some exceptions which was not addressed in the discussion so far.

Response 5: Thank you for highlighting this. The unexpected findings/exceptions have been discussed thus (Lines 402–409);

“Notably, the study shows unexpected lower rates of modern contraceptive use among women with higher compared to secondary education, in the richer compared to middle wealth quintile, and parity ≥6 compared to parity 4–6. Recognising the potential influence of study limitations, these findings suggest that women with higher educational attainment or from wealthier households may prioritise career or personal development over immediate reproductive needs. Also, women with more children may experience "fertility inertia" or less likely to seek contraception after achieving their desired family size.”

Comment 6: Regarding the line 336 and next sentences It is very important to prioritize the poorer and less educated people. But simultaneously, country should develop programs for increasing the literacy level as a standing agenda and a cornerstone for development issues. The social policy and social security relevant organization need to develop the relevant programs.

Response 6: Thank you for your suggestion. The statements have been revised (Lines 356–359) thus;

“Simultaneously, the country should focus on increasing literacy and reducing poverty through its development initiatives as well as collaborative efforts with social services organisations, to enhance understanding and empower decision-making in family planning, particularly for these disadvantaged groups [41].”

Comment 7: In the line 354 what is the meaning of positive trend. Does it mean the gap decrease or the figure increase. For me it is vague.

Response 7: Thank you for pointing this out. ‘Positive’ and ‘Negative’ have been replaced with ‘Increasing’ and ‘Decreasing’. The sentence has been revised (Lines 376–378) thus;

“More unclear are the education-related patterns of unmet need for modern methods, as trends across categories showed increasing levels in most regions (6 out of 10), and decreasing or steady in the rest.

Comment 8: In the line 366 In the text, more than 20 years old is a predictor for increasing contraceptive use. As a usual, with increasing the age and going to the older generation, contraceptive use may decrease. It is necessary to have a short explanation on it.

Response 8: Thank you. We have reinforced the discussions with your recommendation, thus (Lines 394–397);

“Estimates showing that women aged 20 to 29 and 30 to 39 are at least twice more likely to use modern contraceptives compared to 15 to 19-year-olds aligns with literature, which indicates that contraceptive use trends increase with age until they begin to decline in older generations [23, 44].”

Comment 9: In the line 376 and next sentences, uncovering disparities and its reasons will help policymakers to develop a plan toward SDG goals. It will show that different region in the country needs different approaches and intervention.

Response 9: Thank you very much. Your suggestion has been incorporated, thus (Lines 422–424);

“The findings should enable Cameroon to adopt varied approaches for family planning implementation and expansion, tailored to the disparities identified in each region.”

Comment 10: In the conclusion, it is necessary to think about multi-sectoral cooperation for improvement of literacy rate, social support schemes, provision of services with subsidy, improvement of access ...

Response 10: The conclusion has been reinforced with your suggestion, thus (Lines 448–451);

“To effectively address these disparities, multi-sectoral cooperation is essential. This includes initiatives aimed at improving literacy rates, implementing social support schemes, providing subsidised services, and enhancing access to family planning resources.”

Reviewer #2:

Comment: There is no substantial comment to be forwarded, How ever, I recommend the abstract to be rewritten to give a brief and explained background information regarding the topic "interest socioeconomic assessment of family planning levels". Additionally, the conclusion section should be revised to make it focused on the major results.

Response: Thank you for your valuable feedback. The abstract background has been revised to demonstrate our "interest socioeconomic assessment of family planning levels" (Lines 27–31).

---

## [Decision Letter · Decision Letter 1]

2 Dec 2024

PONE-D-24-06584R1A subnational socioeconomic assessment of family planning levels, projections, and disparities among married women of reproductive age in CameroonPLOS ONE

Dear Dr. Nsashiyi,

Thank you for submitting your manuscript to PLOS ONE. After careful consideration, we feel that it has merit but does not fully meet PLOS ONE’s publication criteria as it currently stands. Therefore, we invite you to submit a revised version of the manuscript that addresses the points raised during the review process.

We look forward to receiving your revised manuscript.

Kind regards,

Yibeltal Alemu Bekele, MpH

Academic Editor

PLOS ONE

Journal Requirements:

Reviewers' comments:

Reviewer's Responses to Questions

**Comments to the Author**

1. If the authors have adequately addressed your comments raised in a previous round of review and you feel that this manuscript is now acceptable for publication, you may indicate that here to bypass the “Comments to the Author” section, enter your conflict of interest statement in the “Confidential to Editor” section, and submit your "Accept" recommendation.

Reviewer #1: (No Response)

2. Is the manuscript technically sound, and do the data support the conclusions?

Reviewer #1: Yes

3. Has the statistical analysis been performed appropriately and rigorously? 

Reviewer #1: Yes

4. Have the authors made all data underlying the findings in their manuscript fully available?

Reviewer #1: Yes

5. Is the manuscript presented in an intelligible fashion and written in standard English?

Reviewer #1: Yes

6. Review Comments to the Author

Reviewer #1: Thanks. The comments were mostly addressed. I couldn't find the answer to one comment which was related to the average of marriage among the community. You already mentioned it was addressed but, I couldn't find it.

7. PLOS authors have the option to publish the peer review history of their article (what does this mean? ). If published, this will include your full peer review and any attached files.

**Do you want your identity to be public for this peer review?** For information about this choice, including consent withdrawal, please see our Privacy Policy .

Reviewer #1: **Yes: ** Mohammad Eslami

---

## [Author Response · Author response to Decision Letter 1]

27 Dec 2024

Response to Reviewers

Journal Requirements:

Response:

Thank you for your guidance regarding the reference list. We have thoroughly cross-checked the reference list to ensure that it is complete and accurate. No new references were added; however, we have made several edits for consistency, including the addition of punctuation marks (full stops ‘.’), proper spacing, and the inclusion of ‘https://’ before the digital object identifier (DOI) for each article.

We confirm that there are no retracted papers in our reference list.

Reviewer #1: Thanks. The comments were mostly addressed. I couldn't find the answer to one comment which was related to the average of marriage among the community. You already mentioned it was addressed but, I couldn't find it.

Thank you for your comment.

We have included the mean age of the respondents (i.e., married women of reproductive age) in the descriptive results, as the average age of marriage among the community was not captured in the dataset we utilized. Additionally, we were unable to locate reliable or scientifically sound sources that provide data on the average age of marriage in Cameroon.

To ensure clarity, we have explicitly stated the average age of the respondents in the manuscript. Specifically, Lines 266-268 have been revised to read:

“The descriptive results (Table 3) indicate that among the 1,513 respondents, the average age was 30.8 years (SD ±7.9), with a considerable proportion falling within the 20–29 (38.6%) and 30–39 (37.9%) age groups, while only 7.7% were aged 15–19.”

We appreciate your valuable feedback and hope this revision addresses your concerns.

---

## [Editor Report · Decision Letter 2]

21 Jan 2025

A subnational socioeconomic assessment of family planning levels, projections, and disparities among married women of reproductive age in Cameroon

PONE-D-24-06584R2

Dear Dr. Raïssa Shiyghan Nsashiyi,

We’re pleased to inform you that your manuscript has been judged scientifically suitable for publication and will be formally accepted for publication once it meets all outstanding technical requirements.

Kind regards,

Yibeltal Alemu Bekele, MpH

Academic Editor

PLOS ONE

---

## [Editor Report · Acceptance letter]

PONE-D-24-06584R2

PLOS ONE

Dear Dr. Nsashiyi,

I'm pleased to inform you that your manuscript has been deemed suitable for publication in PLOS ONE. Congratulations! Your manuscript is now being handed over to our production team.

Kind regards,

on behalf of

Mr. Yibeltal Alemu Bekele

Academic Editor

PLOS ONE